# Muscular Ultrasonography in Morphofunctional Assessment of Patients with Oncological Pathology at Risk of Malnutrition

**DOI:** 10.3390/nu14081573

**Published:** 2022-04-10

**Authors:** Juan J. López-Gómez, Katia Benito-Sendín Plaar, Olatz Izaola-Jauregui, David Primo-Martín, Emilia Gómez-Hoyos, Beatriz Torres-Torres, Daniel A. De Luis-Román

**Affiliations:** 1Servicio de Endocrinología y Nutrición, Hospital Clínico Universitario de Valladolid, 47003 Valladolid, Spain; katiaplaar@hotmail.com (K.B.-S.P.); olatzizaola@yahoo.es (O.I.-J.); dprimoma@saludcastillayleon.es (D.P.-M.); emiliagomezhoyos@gmail.com (E.G.-H.); beatriztorrestorres@hotmail.com (B.T.-T.); dadluis@yahoo.es (D.A.D.L.-R.); 2Centro de Investigación en Endocrinología y Nutrición (IENVA), Universidad de Valladolid, 47002 Valladolid, Spain

**Keywords:** body composition, sarcopenia, oncological patient, muscular ultrasonography, impedanciometry

## Abstract

*Background:* Muscular ultrasonography is a technique that allows assessing the amount and quality of muscle in a specific body region. The aim of the study was to compare the value of muscle ultrasonography in diagnosis of malnutrition with techniques such as anthropometry, handgrip strength and impedanciometry in patients with oncological pathology. *Methods:* Cross-sectional study in 43 patients with oncological pathology and high nutritional risk. Classical anthropometry (body mass index (BMI), arm circumference (AC), calf circumference (CC) and estimated appendicular muscle mass index (ASMI)) was performed. Body composition was measured with impedanciometry (BIA), phase angle (PA) and fat-free mass index (FFMI) and muscle ultrasonography of quadriceps rectus femoris (muscle area (MARA) and circumference (MCR) in section transverse). Malnutrition was diagnosed using the GLIM criteria and sarcopenia was assessed using EWGSOP2 criteria. *Results:* The mean age was 68.26 years (±11.88 years). In total, 23/20 of the patients were men/women. The BMI was 23.51 (4.75) kg/m^2^. The ASMI was 6.40 (1.86) kg/m^2^. The MARA was 3.31 cm^2^ in ultrasonography. In impedanciometry, phase angle was 4.91 (0.75)°; the FFMI was 17.01 kg/m^2^ (±2.65 kg/m^2^). A positive correlation was observed between the MARA with anthropometric measurements (AC: r = 0.39, *p* = 0.009; CC: r = 0.44, *p* < 0.01; ASMI: r = 0.47, *p* < 0.001); and with BIA (FFMI: r = 0.48, *p* < 0.01 and PA: r = 0.45, *p* < 0.001). Differences were observed when comparing the MARA based on the diagnosis of sarcopenia (Sarcopenia: 2.47 cm^2^ (±0.54 cm^2^); no sarcopenia: 3.65 cm^2^ (±1.34 cm^2^); *p* = 0.02). *Conclusions:* Muscle ultrasonography correlates with body composition measurement techniques such as BIA and anthropometry in patients with cancer.

## 1. Introduction

Disease-related malnutrition (DRM) is a highly frequent pathology with a prevalence of 60% in hospitalized patients with chronic diseases [1]. Up to 70% of patients maintain a situation of malnutrition at discharge. This malnutrition is more striking in elderly patients and is closely related to sarcopenia, another highly prevalent condition in patients with chronic disease [1].

The patient with oncological pathology of any type presents an increased risk of suffering malnutrition. It has been observed that between 15 and 40% of cancer patients present some degree of malnutrition at diagnosis. This condition worsens with the progression of the disease, with 80% of patients suffering malnutrition in advanced stages [2].

Malnutrition in cancer patients can lead to complications such as reduced tolerance to treatment (surgical, chemotherapy or radiotherapy), increased length of stay, increased costs and increased morbidity and mortality associated with the disease [3,4].

Early diagnosis and treatment of poor nutritional status can positively influence the evolution of the pathology. The identification of patients at risk and the early start of nutritional support has been associated with an improvement in the rate of complications, reducing the length of stay [5,6] and improving quality of life [7].

The diagnosis of malnutrition is difficult because it does not depend only on the weight at a given time, but also on its evolution and the pathological situation that underlie weight loss [8]. Classically, the body mass index (BMI) has been used of the patient’s nutritional status, but this measure is not the most appropriate and has evident limitations in different pathologies [9]. These diseases can produce and increase in fat mass or body water without observing a weight loss [10]. Therefore, the clinical use of body composition measurements is essential for adequate assessment of this malnutrition, especially in the evaluation of muscle mass.

In cancer patients, the cachexia is a multifactorial syndrome that involves multiple factors (inflammatory, reduced intake, treatment damage…) and conditions a continuous loss of muscle mass. This syndrome is characterized by a loss of 5% of weight in the last 6 months, body mass index less than 20 kg/m^2^ and any weight loss more than 2%; or appendicular skeletal mass index compatible with sarcopenia and any weight loss more than 2%. The presence of this disease is related with more complications and poorer outcomes in cancer patients. Body composition and the detection of loss of muscle mass is important to early diagnose of this entity [6].

There are techniques for determining body composition different to the methods mainly used in research due to their difficulty of access and performance, such as air displacement plethysmography, in vivo neutron activation analysis, isotope dilution and total body K count total. Specific imaging techniques (magnetic resonance imaging, computerized tomography (TC) or dual energy X-ray absorptiometry (DEXA)) are accessible in hospitals but are not used in routine clinical practice due to logistical difficulties. Finally, there are methods used with more accessibility and easy implantation in routine clinical practice as classical anthropometry and bioelectrical impedanciometry (BIA) [11]. Recently, the use of muscle ultrasonography has shown a new dynamic alternative in the quantitative and qualitative assessment of muscle mass, as we can see in the studies from Hernandez-Socorro et al. [12]

Muscle ultrasonography is a technique that allows us to assess muscle quantity and quality in a specific region of the body. This technique can take a little time and can be conducted in a consulting room or at a hospital bedside [13]. The main problems are that scientific evidence is poor, there are many potential areas to measure, and it is not clear if the loss of muscle in a specific area correlates with the loss of overall muscle mass. On the other hand, most of the ultrasonography studies have been performed in healthy elderly people; therefore, its role in DRM remains to be validated [14].

In this background, the use of new non-invasive methods to measure body composition, such as bioimpedance and muscle ultrasonography, can help us to diagnose and adapt nutritional treatment appropriately.

The purpose of this study was to compare the muscle ultrasonography with usual techniques such as handgrip strength and BIA in patients with oncological pathology at risk of malnutrition.

## 2. Materials and Methods

### 2.1. Design

This is an open, cross-sectional observational study to evaluate the nutritional status of patients with oncological pathology referred to a Clinical Nutrition Unit.

The study was in compliance with the Declaration of Helsinki 1964 (last update 2013). All procedures were approved by Institutional Review Board (IRB) of East Area of Valladolid (Castilla y León, Spain) under code PI 20-1967 and the next resolution: “Considerando que el Proyecto contempla los Convenios y Normas establecidos en la legislación española en el ámbito de la investigación biomédica, la protección de datos de carácter personal y la bioética, se hace constar el informe favorable y la aceptación del Comité de Ética de la Investigación con Medicamentos Área de Salud Valladolid Este para que sea llevado a efecto dicho Proyecto de Investigación”.

### 2.2. Study Subjects

The study was developed in patients with oncological pathology from the East Valladolid Area referred to Clinical Nutrition Unit for nutritional assessment.

The patient inclusion criteria were: (i) patients with oncological disease at risk of malnutrition; and (ii) age over 18 years. The exclusion criteria were: (i) decompensated liver disease; (ii) chronic kidney disease stage IV or higher; (iii) inability to walk; and (iv) non-signing of the informed consent by the patient.

After signing the informed consent and the inclusion of the patient in the study, an exhaustive anamnesis was carried out on affiliation data, personal history, evolution of the disease and nutritional history. With the data obtained, an initial descriptive statistical analysis of the prevalence and nutritional status of the patients was carried out, comparing the different techniques for evaluating body composition.

### 2.3. Study Variables

An exhaustive anamnesis was carried out on affiliation data, personal history, evolution of the disease and nutritional history. Anthropometry, BIA measurement, hand-grip strength and muscle ultrasound evaluation were performed. Analysis of nutritional parameters was developed according to usual clinical practice.

#### 2.3.1. Clinical Variables

The following were measured: age (years); gender (male/female); systolic and diastolic blood pressure (mmHg). The presence of diabetes mellitus and its type and presence of concomitant pathologies were also checked.

#### 2.3.2. Anthropometric Variables

The anthropometric variables measured were weight (kg); height (meters); body mass index (BMI) (weight/height × height) (kg/m^2^); arm circumference (AC); and calf circumference (CC) [15].

Appendicular skeletal muscle index (ASMI) was estimated using the formula:−10,427 + (CC(cm) × 0.768) − (age(years) × 0.029) + (sex × 7523)/(height(cm) × height(cm))

This formula was made using data from the NHANES study between 1999 and 2006 [16]. EWGSOP2 diagnostic criteria of sarcopenia for low muscle mass (ASMI < 7 kg/m^2^ in and ASMI < 5.5 kg/m^2^ in women) were used [17].

#### 2.3.3. Muscle Strength

Handgrip strength (JAMAR^®^ dynamometer): non-dominant handgrip strength was performed with the patient seated and the arm at a right angle to the forearm. Handgrip strength was measured three times in non-dominant arm; the average of these three measurements was calculated.

The diagnostic criteria for sarcopenia proposed by the European Working Group on sarcopenia in older people [16] were used to assess decreased low muscular strength (<27 kg in men and <16 kg in women).

#### 2.3.4. Body Composition

Bioimpedanciometry (BIA 101 Anniversary; EFG Akern): The BIA was performed between 8:00 and 9:15 h, after an overnight fast and after a time of 15 min in the supine position. The BIA measured the geometrical components of impedance (Z), resistance (R) and the capacitance component (X). The PhA is derived for the next equation PhA = (X/R) × (180°/π). The BIA provided data regarding fat mass (FM), fat-free mass (FFM), skeletal muscle mass (SMM), fat-free mass index (FFMI) and percentage of skeletal muscle mass (%MM) [18] (EFG BIA 101 Anniversary, Akern, It). All these data are based on raw electrical data from BIA [18].

Muscle ultrasonography of the quadriceps rectus femoris (QRF) of the non-dominant lower extremity with a 10 to 12 MHz probe and a multifrequency linear matrix (Mindray Z60, Madrid, Spain) was performed in all subjects (patient in supine position). The probe was aligned perpendicular to the longitudinal and transverse axis of the non-dominant QRF. The evaluation was performed without compression at the level of the lower third from the superior pole of the patella and the anterior superior iliac spine, measuring the anteroposterior muscle thickness, circumference and cross-sectional area [18]. The ultrasonography was made by the same person who was formed in this technique previously. The areas of measurement were standardized. These actions were made in order to reduce intra- and interoperator variability. The measurements made using this technique were: muscle area (cm^2^) (MARA) and the index of the muscle to height (cm^2^/m^2^) (MARAI); muscle circumference (cm) (CMR) and the index circumference to height (cm/m^2^); and transverse subcutaneous adipose tissue (cm) (SCAT), transverse muscle thickness (cm) and the index that relates both (SCAT/muscle thickness) (MAI) [18].

#### 2.3.5. Diagnosis of Malnutrition and Sarcopenia

The diagnosis of malnutrition was made using the criteria of the Global Leadership Initiative on Malnutrition (GLIM), using the FFMI measured by BIA as a variable for assessing muscle deterioration (a reduction in muscle mass was considered FFMI < 17 kg/m^2^ in men and <15 kg/m^2^ in women) [8]. Furthermore, the diagnosis of sarcopenia was made according to the revised European Working Group on Sarcopenia in Older People (EWGSOP2) sarcopenia criteria, using ASMI estimated by calf circumference as a measure of decreased muscle mass and measure of handgrip strength as decreased muscle strength [17].

### 2.4. Statistical Analysis

The data was stored in a database of the statistical package SPSS 23.0 (SPSS Inc., Chicago, IL, USA). A normality analysis of continuous variables was performed with the Kolmogorov–Smirnov test. Continuous variables were expressed as mean (standard deviation). The difference in means between parametric variables was analyzed with the unpaired and paired t-Student, and the non-parametric variables with the Mann–Whitney U-test and the Kruskal–Wallis K-test. A Pearson’s correlation test was used for the comparison of continuous variables. A *p*-value less than 0.05 was considered a significant difference.

## 3. Results

### 3.1. Subject Characteristics

43 patients were analyzed. They had a mean age of 68.26 years (±11.88 years). A total of 23 (53.5%) patients were men.

The distribution of the type of oncological pathology is shown in Figure 1.

The values of the morphofunctional study showed differences according to the sex in most of the variables except in ultrasonography measures adjusted for height, in the muscle mass/adipose tissue index (MAI) and in the phase angle (Table 1).

### 3.2. Correlation between Body Composition Assessment Techniques

The correlation analysis between parameters measured by muscle ultrasonography and the different morphofunctional assessment values are shown in Table 2.

#### 3.2.1. Anthropometry

No correlation was observed between the measurements used in muscle ultrasonography and BMI (Table 2).

When we evaluated anthropometry techniques (arm circumference, calf circumference and ASMI estimated by calf circumference), a correlation was observed with the quadriceps rectus femoris area (MARA) (r = 0.47; *p* < 0.001) (Figure 2a) being more robust when using the index of MARA by height (r = 0.57; *p* < 0.001) (Figure 2b). The QRC circumference (RMC) only showed correlation with estimated ASMI (Table 2).

#### 3.2.2. Body Composition

When we measured the electrical values of the impedanciometry, a high correlation of phase angle with the MARA (r = 0.39; *p* < 0.001) (Figure 3a) and the MARAI (r = 0.41; *p* < 0.001) (Figure 3b) was observed. On the other hand, a negative correlation of these parameters was observed with bioelectric resistance (MARA: r = −0.48; *p* < 0.001; MARAI: r = −0.46; *p* < 0.001) (Figure 3c,d).

A correlation of the FFMI with the muscle area (r = 0.48; *p* < 0.001) (Figure 4a) and circumference (r = 0.45; *p* < 0.001) (Figure 4b) measured by ultrasonography was observed. Likewise, a negative correlation was observed between the subcutaneous adipose tissue of muscle mass and a correlation with the muscular adipose index estimated by ultrasonography (Table 2).

#### 3.2.3. Muscle Strength

Non-dominant handgrip strength showed a correlation with the absolute measurements of MARA (r = 0.45; *p* < 0.001) (Figure 5a) and MCR (r = 0.32; *p* = 0.009) (Figure 5b). On the other hand, a correlation was also observed with the muscle/fat mass index measured by ultrasonography (Table 2).

### 3.3. Diagnosis of Malnutrition with GLIM

The malnutrition rate in the sample according to the GLIM criteria, showed that 36 (83.7%) patients presented malnutrition, and 19 (44.2%) patients had severe malnutrition.

No differences were observed in muscle ultrasonography and bioimpedanciometry, in terms of muscle ultrasonography parameters in patients diagnosed with malnutrition, according to GLIM criteria compared to those who did not suffer from it (Table 3). They were not observed in relation to the severity of malnutrition diagnosed by GLIM criteria (Table 3).

### 3.4. Diagnosis of Sarcopenia

Regarding the diagnosis of sarcopenia, it was observed that 13 (30.2%) patients presented criteria of low strength and low muscle mass according to the EWGSOP2 criteria.

When analyzing the muscle ultrasonography and bioimpedanciometry measurement data based on the diagnosis of sarcopenia, it was observed that patients with sarcopenia according to the EWGSOP2 criteria had lower values of MARA and MCR in addition to higher levels of resistance, BMCI and FFMI in bioimpedanciometry (Table 4).

## 4. Discussion

Muscle ultrasound for the assessment of body composition is a technique that can provide us with quick information of patients at nutritional risk. In this study carried out in patients with oncological pathology, this technique has shown a correlation with body mass determination techniques such as calf circumference, in addition to body composition parameters measured by impedance measurement and muscle strength measured by dynamometry. However, no differences were observed when the diagnosis of malnutrition was evaluated using the GLIM criteria, although a difference was observed when the diagnosis of sarcopenia was applied using the EWGSOP2 criteria.

The nutritional assessment of cancer patients cannot be carried out solely through anthropometric measurements but should be completed with measurements of body composition and functionality to carry out a more adequate diagnosis and monitoring [18]. Classic measurements such as BMI present a lack of information, as we saw no differences found in this parameter when we compared it to ultrasonography, and no correlation was observed with this new technique. Nevertheless, BMI is the parameter most used in diagnosis with GLIM criteria, as Correia et al. showed in a recent study [19]. We need to use more accurate measurements such as ultrasonography to detect malnutrition.

It is usually necessary to consider different scales based on sex as defined both in the EWGSOP2 criteria [17] for sarcopenia and in the GLIM criteria [8] for malnutrition. There are constitutional differences between men and women, as well as taking into account age. In the patients analyzed, differences were observed between both sexes in all absolute parameters related to mass, body composition and muscle strength.

In the analyzed sample, it was observed that some anthropometric parameters such as body mass index or arm and calf perimeters were within a normal range. It was observed that data was altered by applying an estimation of muscle mass by means of a formula based on NHANES study [16] in other studies in which cancer patients were analyzed, such as that by Sánchez-Torralvo et al. [15] or Gort-VanDijk et al. [20]. Body mass index and classical anthropometric parameters were found to be within the normal range, but when performing an analysis using computerized tomography (CT) [15] and impedanciometry measurement (BIA), altered parameters of body composition could be observed [20].

The values obtained in the handgrip strength were like those of the series of patients with oncological pathology by Contreras-Bolívar et al. In this study, the obtained measurements of handgrip strength in the total sample and differenced according to sex were similar to those of our series [21].

Nutritional ultrasonography is a technique under development and there are no standardized locations and cut-off points yet. The SARCUS group proposes a series of locations in different muscle groups and parameters related to muscle structure, muscle quality, contraction or circulation in it [22]. In this study, the structural component of the muscle and lipid mass of the location of the lower third of the QRC was evaluated. The decision to use this location was related to being an area involved in body function, being easily reproducible and detectable by different observers [22].

Regarding the general description of the ultrasonography study, differences between the sexes were observed in the muscular component and in the premuscular adipose component. These differences at the muscle level between sexes are observed in other studies carried out on the elderly population, considering muscle thickness as a parameter [23]. For this reason, an index was proposed that would relate adipose tissue and muscle thickness, which did not show differences between sexes. Unlike these studies, we used circumference and muscle area [24].

A clear correlation was observed with the calf circumference and the estimated ASMI with ultrasonography parameters. The relationship established between these parameters seems normal due to the high correlation that calf circumference shows with muscle mass in most studies [17,25]. However, this parameter may be influenced by the situation of inflammation and nonclinical edema that these patients may present in the context of their underlying pathology and chemotherapy treatment [25]. The evaluation of arm circumference is a less valuable parameter due to the variability in its fat component and the differences in its measurement; this may justify the absence of correlation at this level in some parameters measured by muscle ultrasonography [20]. On the other hand, in the case of the BMI, the low utility is observed in normal ranges since even with body mass indexed in normal ranges, striking alterations are observed in the muscle compartment without correlation with the ultrasound [26].

Perhaps the most striking data observed are the correlations obtained with impedanciometry measurement, since a correlation was observed in the indicators related to cell mass, a negative correlation with resistance and a correlation with the phase angle. This may be related to the low-fat component of these patients and the relationship with fat-free mass, which was confirmed by evaluating the FFMI. The ultrasonography variable that showed the high correlation was the muscle area and its index related to height. On the other hand, the index of fat-free mass and the index of muscle mass showed a good correlation with muscular parameters of the ultrasound. These data show us that in this group of patients, the use of muscle ultrasonography, especially parameters based on muscle area, can help us to assess the total amount of potentially active cell mass. In addition, the relationship with the phase angle can guide us towards the combined use of these two techniques available in routine clinical practice to monitoring the nutritional status in cancer patients [27].

The diagnosis of malnutrition was analyzed using the GLIM criteria, and a high prevalence of this pathology was observed among the patients with oncological pathology. This prevalence is higher than those in other series such as Gascón-Ruiz et al., who observed 46.7% of patients with malnutrition compared to 83.7% found in the studied sample [28]. This situation can be related to the selection of patients, given that this was carried out among patients referred to the clinical nutrition consultation with a greater deterioration in nutritional status than those that could be located on the total number of patients with oncological pathology in routine clinical practice in oncology.

No differences in muscle mass ultrasonography were observed between malnourished and non-malnourished patients. This situation can be related, in the first place, to the selection of patients previously outlined and the low sample size of patients without malnutrition to make the comparison. On the other hand, the use of FFMI values for the diagnosis of muscle mass loss may be interfered by the inflammatory situation of patients with a greater degree of malnutrition and a greater accumulation of total body water that would interfere with measurement [29]. When evaluating the deterioration of muscle mass, it would be more interesting to use electrical parameters such as phase angle, but there are no standards at this level in the definition of the GLIM criteria [30].

Phase angle has shown a relationship with poorer outcomes in cancer patients, as in the study by Axelsson et al. in head and neck cancer [31] or Paiva et al. in patients with cancer who were receiving chemotherapy [32]. Nevertheless, recent studies have shown that is not possible use phase angle as an accurate indicator of malnutrition [33]. In our study, we did not find differences in phase angle between patients with malnutrition or no malnutrition, or between patients with sarcopenia or no sarcopenia. This could be related with changes in corporal water related with inflammatory state in these patients, or with the fact that all patients were at risk of malnutrition when they were referred to the Clinical Nutrition Unit. Ultrasonography parameters were more accurate in these differences, especially in sarcopenia diagnosis.

It was observed that one third of the patients presented EGWSOP2 criteria for sarcopenia diagnosis when this diagnosis was evaluated in its different spheres [17]. A somewhat lower prevalence of sarcopenia was observed than in other studies, such as the systematic review by Hanna et al., which showed 58% of patients, although they referred only to low muscle mass [34]. In the same way, the review by Catikkas et al. observed prevalences between 42.8% and 72%, higher than those of the studied sample [35]. The proposed prevalence is like that described in other series such as the meta-analysis in hematological patients by Surov et al. with 39.1% [36], or that of Trejo-Ávila et al. with a 37% prevalence [37].

However, a significant decrease in muscle mass was observed in those patients diagnosed with sarcopenia. This is important when we consider the influences of different inflammatory parameters on muscle mass and its functionality [21]. For this reason, techniques that allow us to evaluate the muscle directly can be very useful. This is important when we observe the correlation between the amount of muscle defined by ultrasonography and muscle strength. The evaluation of certain quantitative and qualitative parameters of the ultrasound can inform us of the functionality of the muscle in addition to its quantity, and the places of measurement as proposed by the SARCUS group [21]. We need to define and standardize quality parameters in ultrasonography to relate changes at this point with handgrip strength.

The limitations of the study were: (i) The small sample size and the variability that exists between the different oncological pathologies analyzed. The main reason of this small sample size is the selection of patients with a single observer to minimize the variability. (ii) On the other hand, since there were no defined standard cut-off points, we were unable to make a comparison beyond the relationship between the different techniques. (iii) Finally, the use of techniques to qualitatively assess the muscle could have provided us with more information in order to evaluate the influence of inflammation on muscle morphology and its relationship with functionality.

The strengths of this study lie in: (i) The approach of body composition measurement techniques and evaluating their relationship with classical techniques in complex patients such as cancer patients. (ii) This analysis shows us a part of the operation of these new techniques in real clinical practice and allows us to generate hypotheses for the design of studies on their daily use in a more accurate way.

## 5. Conclusions

In the patient with oncological pathology and nutritional risk, muscle mass determined by ultrasonography correlated with body composition (classical anthropometry and impedanciometry). Likewise, a correlation was observed between muscle mass measured by ultrasonography and muscle strength determined with handgrip strength.

This study shows ultrasonography in cancer patients as a safe and cost-effective diagnosis tool that can be compared to other techniques used in clinical routine practice. This tool may help us to complete malnutrition diagnosis in a patient with many factors that influence body composition. On the other hand, this technique can facilitate the surveillance of the medical nutritional treatment.

It is necessary to standardize anatomical landmarks and measure points for all muscles/muscle groups. Despite this need for standardization of the measurement technique, muscle ultrasound has important advantages such as low economic cost, zero exposure to radiation, non-invasive technique and a short period of time for exploration.

It is very important to change the vision of nutritional assessment from a static point of view to a morphofunctional assessment. This change must be based on the use of multiple techniques that help us to know the body composition (quantity and quality of different tissues), muscle strength, muscle functionality and eating assessment.

## Figures and Tables

**Figure 1 nutrients-14-01573-f001:**
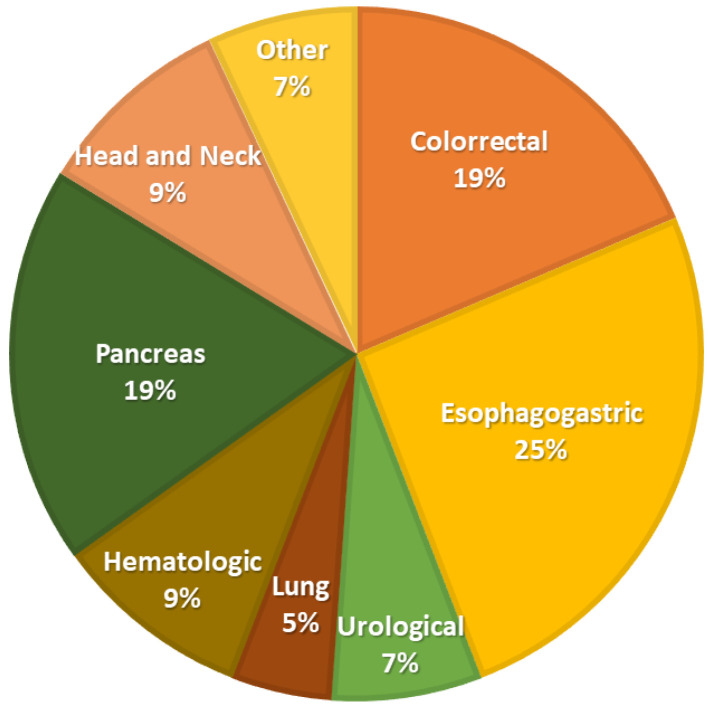
Distribution of oncological disease.

**Figure 2 nutrients-14-01573-f002:**
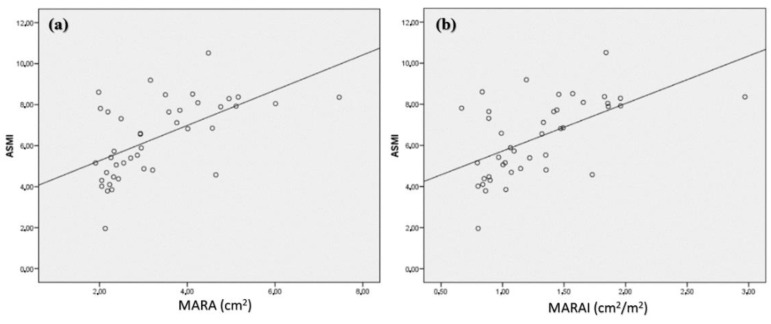
Correlation between estimated ASMI with MARA (**a**) and MARAI (**b**). MARA: muscular area rectus anterior, MARAI: muscular area rectus anterior index.

**Figure 3 nutrients-14-01573-f003:**
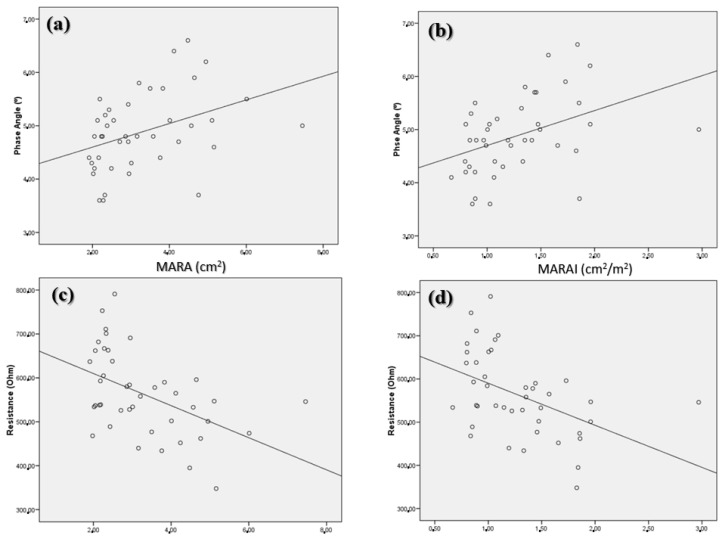
Correlation between phase angle (**a**,**b**) and electrical resistance (**c**,**d**) with MARA (**a**,**c**) and MARAI (**b**,**d**). MARA: muscular area rectus anterior, MARAI: muscular area rectus anterior index.

**Figure 4 nutrients-14-01573-f004:**
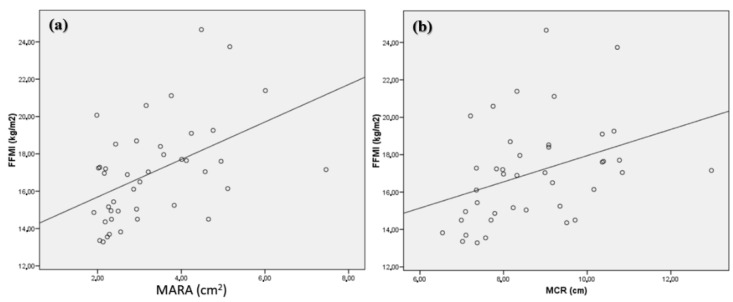
Correlation between FFMI and electrical resistance with MARA (**a**) and MCR (**b**). FFMI: fat-free mass index; MARA: muscular area rectus anterior, MCR: muscular circumference rectus.

**Figure 5 nutrients-14-01573-f005:**
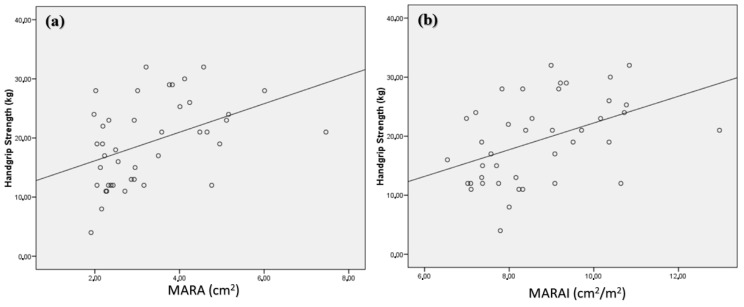
Correlation between handgrip strength and electrical resistance with MARA (**a**) and MCR (**b**). MARA: muscular area rectus anterior, MCR: muscular circumference rectus.

**Table 1 nutrients-14-01573-t001:** Differences in morphofunctional assessment between sex.

	Total	Men N = 23	Women N = 20	*p*-Value
CLASSICAL ANTHROPOMETRY				
% weight loss	10.37 (±8.42)	10.9 (±7.11)	9.57 (±10.15)	0.626
BMI (kg/m^2^)	23.51 (±4.75)	24.98 (±3.89)	21.82 (±5.17)	0.028
Arm circumference (cm)	24.32 (±3.73)	25.66 (±2.87)	22.79 (±4.08)	0.010
Calf circumference (cm)	32.48 (±3.4)	33.78 (±3.38)	30.99 (±2.82)	0.006
ASMI estimated (kg/m^2^)	6.40 (±1.86)	7.84 (±1.06)	4.74 (±0.98)	<0.001
MUSCULAR STRENGTH				
Handgrip strength	19.73 (±7.69)	24.01 (±6.36)	14.8 (±6.01)	<0.001
MUSCULAR ULTRASONOGRAPHY RECTUS ANTERIOR FEMORIS				
SCAT (cm)	0.61 (±0.33)	0.46 (±0.22)	0.71 (±0.34)	0.008
MAI	8.43 (±6.96)	9.94 (±8.13)	6.85 (±5.23)	0.158
MARA (cm^2^)	3.31 (±1.17)	3.97 (±1.34)	2.53 (±0.61)	0.002
MARAI (cm^2^/m^2^)	1.29 (±0.44)	1.48 (±0.51)	1.05 (±0.24)	<0.001
MCR (cm)	8.86 (±1.31)	9.48 (±1.38)	7.86 (±0.88)	<0.001
MCRI (cm/m^2^)	3.49 (±0.53)	3.57 (±0.60)	3.26 (±0.42)	0.068
BIOIMPENDACIOMETRY				
Resistance (W)	561.02 (±96.12)	505.04 (±68.04)	625.39 (±83.10)	<0.001
Reactance (W)	47.97 (±9.37)	44.74 (±8.79)	51.68 (±8.81)	0.019
Phase angle (°)	4.91 (±0.75)	5.07 (±0.79)	4.73 (±0.66)	0.141
BCMI (kg/m^2^)	7.64 (±1.69)	8.39 (±1.58)	6.74 (±1.37)	<0.001
FFMI (kg/m^2^)	17.01 (±2.65)	18.43 (±2.55)	15.37 (±1.68)	<0.001

BMI: body mass index, ASMI: appendicular skeletal muscle index, SCAT: subcutaneous adipose tissue, MAI: muscular adipose index, MARA: muscular area rectus anterior, MARAI: muscular area rectus anterior index, MCR: muscular circumference rectus, MCRI: muscular circumference rectus index, BCMI: body cell mass index, FFMI: fat-free mass index.

**Table 2 nutrients-14-01573-t002:** Correlations of body composition parameters measured by ultrasonography with other morphofunctional assessment parameters (anthropometry, handgrip strength and impedanciometry).

N = 43	MARA (cm^2^)	MARAI (cm^2^/m^2^)	MCR (cm)	MCRI (cm/m^2^)	SCAT (cm)	MAI
BMI (kg/m^2^)	r = 0.26 *p* = 0.092	r = 0.28 *p* = 0.076	r = 0.12 *p* = 0.475	r = 0.14 *p* = 0.388	r = 0.27 *p* = 0.097	r = −0.27 *p* = 0.097
Arm circumference (cm)	*r* = *0.39 ** *p* = *0.011*	*r* = *0.35 ** *p* = *0.023*	r = 0.21 *p* = 0.183	r = 0.09 *p* = 0.545	r = 0.162 *p* = 0.334	r = −0.16 *p* = 0.334
Calf circumference (cm)	*r* = *0.44 ** *p* = *0.003*	*r* = *0.38 ** *p* = *0.012*	r = 0.21 *p* = 0.190	r = 0.03 *p* = 0.838	r = 0.16 *p* = 0.310	r = −0.04 *p* = 0.820
ASMI (kg/m^2^)	*r* = *0.47 ** *p* < *0.001*	*r* = *0.57 ** *p* < *0.001*	*r* = *0.58 ** *p* < *0.001*	*r* = *0.34 ** *p* = *0.030*	r = 0.18 *p* = 0.273	r = −0.14 *p* = 0.381
Hand grip strength (kg)	*r* = *0.45 ** *p* = *0.007*	*r* = *0.32 ** *p* = *0.037*	*r* = *0.81 ** *p* < *0.001*	*r* = *0.45 ** *p* < *0.001*	r = −0.27 *p* = 0.095	*r* = *0.34 ** *p* = *0.029*
Phase angle (º)	*r* = *0.39 ** *p* = *0.004*	*r* = *0.41 ** *p* < *0.001*	r = 0.26 *p* = 0.099	r = 0.25 *p* = 0.110	r = 0.01 *p* = 0.939	r = 0.06 *p* = 0.710
Resistance (Ω)	*r* = *−0.48* *p* < *0.001*	*r* = *−0.46* *p* < *0.001*	*r* = *−0.54* *p* < *0.001*	*r* = *−0.44* *p* < *0.001*	r = 0.26 *p* = 0.101	r = −0.17 *p* = 0.294
Reactance (Ω)	r = −0.13 *p* = 0.406	r = −0.09 *p* = 0.542	r = −0.26 *p* = 0.100	r = −0.18 *p* = 0.268	r = 0.23 *p* = 0.154	r = −0.11 *p* = 0.493
FFMI (kg/m^2^)	*r* = *0.48 ** *p* < *0.001*	*r* = *0.45 ** *p* < *0.001*	*r* = *0.37 ** *p* = *0.018*	r = 0.26 *p* = 0.100	*p* = 0.29 r = 0.071	r = −0.04 *p* = 0.812
BCMI (kg/m^2^)	*r* = *0.53 ** *p* < *0.001*	*r* = *0.53 ** *p* < *0.001*	*r* = *0.40 ** *p* = *0.010*	*r* = *0.32 ** *p* = *0.047*	r = −0.01 *p* = 0.969	r = 0.03 *p* = 0.846
%MM	*r* = *0.31 ** *p* = *0.049*	r = 0.24 *p* = 0.131	*r* = *0.42 ** *p* < *0.001*	r = 0.21 *p* = 0.192	*r* = *−0.49 ** *p* = <*0.001*	*r* = *0.51 ** *p* < *0.001*

ASMI: appendicular skeletal muscle index, SCAT: subcutaneous adipose tissue, MAI: muscular adipose index, MARA: muscular area rectus anterior, MARAI: muscular area rectus anterior index, MCR: muscular circumference rectus, MCRI: muscular circumference rectus index, BCMI: body cell mass Index, FFMI: fat-free mass index; %MM: percentage muscle mass. ** p*-value < 0.05.

**Table 3 nutrients-14-01573-t003:** Differences between the morphofunctional assessment parameters (ultrasonography and bioimpedanciometry) based on the diagnosis of malnutrition and severe malnutrition using GLIM criteria.

	Malnutrition N = 36	*p*-Value	No Malnutrition N = 7	*p*-Value	Severe Malnutrition N = 19
SCAT (cm)	0.59 (±0.29)	0.403	0.70 (±0.50)	0.293	0.55 (±0.24)
MAI	8.52 (±7.19)	0.857	7.99 (±6.18)	0.570	1.09 (±8.98)
MARA (cm^2^)	3.33 (±1.36)	0.623	3.06 (±0.69)	0.620	3.32 (±1.29)
MARAI (cm^2^/m^2^)	1.29 (±0.49)	0.753	1.23 (±0.23)	0.811	1.27 (±0.48)
MCR (cm)	8.64 (±1.49)	0.600	8.95 (±0.97)	0.813	8.82 (±1.31)
MCRI (cm/m^2^)	3.38 (±0.58)	0.337	3.59 (±0.23)	0.332	3.40 (±0.50)
Resistance (W)	562.99 (±104.11)	0.764	550.86 (±36.17)	0.985	550.10 (±100.05)
Reactance (W)	47.46 (±9.82)	0.428	50.57 (±6.55)	0.337	46.53 (±9.95)
Phase angle (°)	4.85 (±0.77)	0.232	5.23 (±0.53)	0.312	4.87 (±0.85)
BMCI (kg/m^2^)	7.57 (±1.82)	0.548	8 (±0.82)	0.690	7.68 (±1.97)
FFMI (kg/m^2^)	17.05 (±2.86)	0.818	16.79 (±1.28)	0.764	17.16 (±3.11)

SCAT: subcutaneous adipose tissue, MAI: Muscular Adipose Index, MARA: muscular area rectus anterior, MARAI: muscular area rectus anterior index, MCR: muscular circumference rectus, MCRI: muscular circumference rectus index, BCMI: body cell mass index, FFMI: fat-free mass index.

**Table 4 nutrients-14-01573-t004:** Differences between the morphofunctional assessment parameters (ultrasonography and bioimpedanciometry) based on the diagnosis of sarcopenia using EGWGSOP2 criteria.

	Sarcopenia N = 13	No Sarcopenia N = 30	*p*-Value
SCAT (cm)	0.61 (±0.37)	0.60 (±0.32)	0.250
MAI	7.89 (±5.44)	8.68 (±7.65)	0.101
MARA (cm^2^)	2.47 (±0.54)	3.65 (±1.34)	0.0041
MARAI (cm^2^/m^2^)	0.99 (±1.90)	1.41 (±0.49)	0.0061
MCR (cm)	7.94 (±1.11)	9.04 (±1.42)	0.0071
MCRI (cm/m^2^)	3.20 (±0.47)	3.52 (±0.55)	<0.001
Resistance (W)	619.76 (±89.27)	535.57 (±88.69)	0.007
Reactance (W)	50.05 (±8.19)	47.07 (±9.83)	0.345
Phase angle (°)	4.62 (±0.54)	5.04 (±0.79)	0.091
BMCI (kg/m^2^)	6.5 (±1.24)	8.1 (±1.65)	0.006
FFMI (kg/m^2^)	15.36 (±1.69)	17.72 (±2.69)	0.004

SCAT: subcutaneous adipose tissue, MAI: Muscular Adipose Index, MARA: muscular area rectus anterior, MARAI: muscular area rectus anterior index, MCR: muscular circumference rectus, MCRI: muscular circumference rectus index, BCMI: body cell mass index, FFMI: fat-free mass index.

## Data Availability

Not applicable.

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
