# Peer review of "Muscular Ultrasonography in Morphofunctional Assessment of Patients with Oncological Pathology at Risk of Malnutrition"

_nutrients, 2022, doi:10.3390/nu14081573_

Round 1

Reviewer 1 Report

The authors conducted a cross sectional study and tried to determine the association between muscular ultrasonography and related body composition measurement techniques to assess the risk of malnutrition among cancer patients.  The author provided sufficient background and addressed the knowledge gap in a logic and scientific manner leading to the research question.  The results are presented in tables and figures, clear and concise in general. The authors discussed the results which were mostly focused on the correlation between the body composition measurement techniques associated with sarcopenia and malnutrition. Although the discussion has touched all the aspects of the results, it could be more organized and focused on main findings. 

Specific comments include:

  • At discussion section, the authors need to determine which is the main focus and what are the major parameters to be focused for discussion. For example, BMI is not an appropriate measurement for body composition, which was not found significance among patients in this study, along with other anthropometric measurements.  Then the focus should be on muscular ultrasonography and its significantly associated parameters.
  • Also at the discussion section, additional discussion is needed to address the hand grip strength and its association with other parameters if any correlation exists or not. One thing needs to be clarified is the correlation between the hand grip strength and the ultrasonography conducted on the quadriceps as the function and structural/composition of the muscle group does not match.
  • The author pointed out one of the limitations is the subject number. A priori power analysis was easy to conduct.  Additional enlightenment on the number of subjects would be very helpful for discussion.
  • Check Line 14 and 16 for appropriate format of parenthesis.
  • Line 38. “…. any type presents and increased risk …” meant to be “… any type presents an increased risk…”?
  • Line 101-111. Consider use complete sentences to combine in one paragraph.
  • Line 114. Consider rewrite the sentence “Three measurements were made and …. Was made”. Please clarify it measured three times or measured at different locations. 

Author Response

Dear reviewers and editorial office:

First, I would like to thank you for the trust placed in our group by reviewing and considering our article.

According to the comments received, we have made a series of corrections in our article that I list below:

Comments and Suggestions for Authors

Reviewer 1:

The authors conducted a cross sectional study and tried to determine the association between muscular ultrasonography and related body composition measurement techniques to assess the risk of malnutrition among cancer patients.  The author provided sufficient background and addressed the knowledge gap in a logic and scientific manner leading to the research question.  The results are presented in tables and figures, clear and concise in general. The authors discussed the results which were mostly focused on the correlation between the body composition measurement techniques associated with sarcopenia and malnutrition. Although the discussion has touched all the aspects of the results, it could be more organized and focused on main findings. 

Specific comments include:

  • At discussion section, the authors need to determine which is the main focus and what are the major parameters to be focused for discussion. For example, BMI is not an appropriate measurement for body composition, which was not found significance among patients in this study, along with other anthropometric measurements.  Then the focus should be on muscular ultrasonography and its significantly associated parameters.
    • We have changed some points in discussion to focused it. We have added a paragraph to discussion to explain limitation of BMI. Line 275-282: “The nutritional assessment of cancer patients cannot be carried out solely through anthropometric measurements but should be completed with measurements of body composition and functionality in order toto carry out a more adequate diagnosis and monitoring [17]. Classic measurements as BMI presents a lack of information as we saw in no differences found in this parameter when we have compared to ultrasonography and no correlation observed with this new technique Nevertheless, BMI is the parameter most used in diagnosis with GLIM criteria as Correia et al has shown in a recent study [18]. We need to use more accurate measurements as ultrasonography to detect malnutrition.”
  • Also at the discussion section, additional discussion is needed to address the hand grip strength and its association with other parameters if any correlation exists or not. One thing needs to be clarified is the correlation between the hand grip strength and the ultrasonography conducted on the quadriceps as the function and structural/composition of the muscle group does not match.
    • We have mentioned previously the correlation between quantitative parameters and handgrip strength. Nevertheless, we don’t use the qualitative parameters of muscle as we don’t have and standardization of these. We have mentioned these points on discussion line 380-386: “For this reason, techniques that allow us to evaluate the muscle directly can be very useful. This is important when we observe the correlation between the amount of muscle defined by ultrasonography and muscle strength. The evaluation of certain quantitative and qualitative parameters of the ultrasound can inform us of the functionality of the muscle in addition to its quantity, and the places of measurement as proposed by the SARCUS group [20]. We need to define and standardize quality parameters in ultrasonography to relate changes at this point with handgrip strength.”
  • The author pointed out one of the limitations is the subject number. A priori power analysis was easy to conduct.  Additional enlightenment on the number of subjects would be very helpful for discussion.
    • We have completed this limitation and explained that in text. Line 388-390: “The main reason of this small sample size is the selection of patients with a single ob-servator to minimize the variability.”
  • Check Line 14 and 16 for appropriate format of parenthesis.
    • We have changed the format of parenthesis.
  • Line 38. “…. any type presents and increased risk …” meant to be “… any type presents an increased risk…”?
    • We have done the change that you suggest on this sentence.
  • Line 101-111. Consider use complete sentences to combine in one paragraph.
    • We have made complete sentences in these two paragraphs. Line 110-117: “2.3.1. Clinical Variables. It has been measured: aAge (years); gender (male/female); systolic and diastolic blood pressure (mmHg). It also has been checked pPresence of diabetes mellitus and its type and. pPresence of concomitant pathologies.  Anthropometric Variables:  The anthropometric variables measured were Wweight (kg); height (meters), body mass index (BMI) (weight/height*height) (kg/m2); arm circumference (AC) and ; calf circumference (CC).”
  • Line 114. Consider rewrite the sentence “Three measurements were made and …. Was made”. Please clarify it measured three times or measured at different locations. 
    • This sentence has been clarified as you ask. Line 124-128: “Handgrip strength (JAMAR® dynamometer) [14]: Non-dominantnon-dominant handgrip strength was performed with the patient seated and the arm at a right angle to the forearm. Handgrip strength was measured three times in non-dominant arm; it was made the average of these three measurements. Three measurements were made and the average of the three measurements was made.”

Reviewer 2 Report

In cancer patients the risk of malnutrition is very high. Patients already at diagnosis, during the first visit with the oncologist, are suffering from malnutrition. This condition only worsens the health condition and can negatively affect the drug response. Promptly identifying this condition with a highly safe method could improve the health conditions of these patients both in terms of quality of life and in response to therapies, furthermore it could be possible to intervene by referring patients to nutrition specialists as well as rehabilitation therapy. I agree with what the authors conclude.

Author Response

Dear reviewers and editorial office:

First, I would like to thank you for the trust placed in our group by reviewing and considering our article.

According to the comments received, we have made a series of corrections in our article that I list below:

In cancer patients the risk of malnutrition is very high. Patients already at diagnosis, during the first visit with the oncologist, are suffering from malnutrition. This condition only worsens the health condition and can negatively affect the drug response. Promptly identifying this condition with a highly safe method could improve the health conditions of these patients both in terms of quality of life and in response to therapies, furthermore it could be possible to intervene by referring patients to nutrition specialists as well as rehabilitation therapy. I agree with what the authors conclude.

  • We are glad that you agree with our conclussions. Thank you very much for the comments.

Reviewer 3 Report

Thank you for the opportunity to review this work. The manuscript under this review concerns principally the comparative of the muscle ultrasonography with usual techniques such as handgrip strength and bioelectrical impedanciomethry in patients with oncological pathology at risk of malnutrition.

The study is interesting and original. However, there are some important concerns that need to be addressed before the manuscript could be considered for publication in Nutrients.

The English must be revised by a native speaker and the writing of the entire manuscript must be improved.

Abstract

The presentation of the abstract needs some clarifications for a better understanding by the reader.

The sections of the abstract; Background, Methods, Results and Conclusions must be presented in italics.

Line 10: The aim should still be in the "background" section and should start with. "The aim of the study was to compare...".

Line 19: Quantitative data when expressed in the abstract and throughout the manuscript, I suggest that they be expressed as: 68.26 years (±1.86 years), as well as 23.51 kg/m2 (±4.75 mg/m2). Thus, throughout the abstract and throughout the manuscript with all the variables presented.

Line 22: p-values ​​should always be presented to three decimal places. Modify it in the abstract and throughout the manuscript.

Introduction

The introduction should have detailed information and provide more exhaustive feedback. Expose and list these different techniques for a better understanding by the reader.

Lines 34-36: Add references that support these sentences that the authors present.

Lines 57-60: What techniques define body composition? The authors must present in detail the existence of the theoretical framework.

Line 61: What does DEXA mean? The first time it occurs in the manuscript, define this acronym. In the same way with all those shown throughout the manuscript.

Lines 64-65: Likewise, I need the authors to support this sentence with a reference. Recently, what do you want to explain with this? References? Relevant studies?

Lines 73-75: I´m agreed with the statement indicated, but I consider that it is an assessment more typical of the discussion section. Please change this sentence. Or is it a hypothesis? Think about this sentence.

Material and Methods

2.1 Design

Lines 83-92: I suggest that this paragraph be changed and presented in subsection 2.2 Study subjects. This information is not included in the study design. Also, the definition of study variables is preferable to present in later subsections, deleting them from the Design subsection.

Lines 90-92: This CEIm is from a Hospital in the Eastern area of ​​Valladolid? It is more specific to present this information from a Hospital Research Ethics Committee with an Institutional Review Board (IRB) instead of a "code number". Declaration of Helsinki; please, describe it according to the last update and the year of creation, for example “The study was in compliance with the Declaration of Helsinki 1964 (last update 2013)”.

Ethical considerations must be rewritten and improved for the manuscript to be considered for publication.

2.2 Study subjects

This subsection should be expanded with the considerations outlined above, improving the information on the inclusion and exclusion criteria, enumerating by i) ii) iii) successively.

2.3 Study variables

The variables are well presented to the reader. Expand the information of the suggested change in subsection 2.1.

2.4 Data analysis

I suggest that this subsection be renamed “Statistical Analysis”

Line 150: The sentence "with an official license from the University of Valladolid" is not required. Delete it.

Results

This section should be improved with the considerations listed above in abstract and below as follows;

First, lines 158-160 should be removed as they are the template hints and should not be shown to reviewers.

3.1. Sample Characteristics

Instead of naming this section as 3.1 Sample Characteristics, I suggest it be renamed as 3.1 Subject Characteristics

If the authors have Figure 1 in color, they can present it, since it offers a better visualization of the results for the reader.

Lines 167-169: Please, in the significant differences, indicate the p-value of the results as indicated in Table 1, these being presented with three decimal places.

3.2 Correlation between body composition assessment techniques

Lines 176-178: It is not necessary for the authors to explain again the analysis performed and presented in Table 2. Rewrite this sentence. It is confusing to the reader.

In Table 2, some results are presented in italics and others are not. Please unify the presentation. p-values, with three decimal places.

3.2.1 Anthropometry

Lines 190-194: Indicate in this paragraph the values ​​p and r and delete them from the figure. In the text, indicate figure 2a and 2b where appropriate, to clarify the presentation of the results.

Line 195: Please, remove this sentence.

3.2.2 Body Composition

Lines 199-202: Indicate the p and r values ​​in the text and delete them from the figures. Indicate the results where they correspond as figure 3a, figure 3b, figure 3c... to facilitate the reader's reading.

Lines 207-210: Same considerations as lines 199-202.

3.2.3 Handgrip Strength

Lines 215-217: Same considerations as above. These sentences are difficult to understand. Are there correlations in both MARA and MCR along with muscle/fat mass index? Re-write this sentence.

3.3 Diagnosis of malnutrition with GLIM

Don't start most sentences with the word "When". It is not aesthetic in English language. Re-write the beginning of the sentences.

Table 3. p-values, please with three decimals.

3.4. Diagnosis of sarcopenia

Line 238: Explain the meaning of EWGSOP2, since it appears numerous times throughout the manuscript.

Change the word criteria instead of "criteria"

Table 4: p-values, please with three decimals.

Discussion

I recommend the authors to rewrite this section for a better understanding. It is poor in meaning that you want to convey.

Avoid starting many of the sentences with the word "When". Use synonyms so as not to be repetitive.

I recommend the authors to rewrite this section for a better understanding. It is poor in meaning that you want to convey. I suggest conducting a deeper systematic review to substantiate the claims in this section.

Lines 271: The Sanchez-Torralvo and Gort-VanDijk studies should be referenced just after the name, for a better understanding of the comparisons made by the authors. You must re-word the sentences between lines 270-274 to organize this information.

Lines 275-277: I suggest re-wording this sentence because the meaning of the Contreras-Bolivar study is not understood; "in total and in differences according to sex"?? What does the author mean with this sentence?

Lines 292-294: Is the main objective of the study necessary to explain it in the middle of the Discussion section? Please consider deleting this sentence or moving it around.

Line 298: In what "most studies"? Please add references to support this statement.

Lines 295-306: This paragraph is devoid of references that support/disagree with the authors' arguments. Please, improve and re-word this paragraph.

Line 329: “first” instead of “firs”.

Line 340: “Hanna et al.”, instead of “Hanna eta al,”

Line 342: Where is the reference for the Catikkas et al. review?

Lines 355-366: On limitations and strengths of the study, please, re-word these paragraphs.

Main limitations or several limitations?

I would place the strengths before the limitations of the study and number them for a better understanding as i) ii) iii)... so on each strength and limitation.

Conclusions

Avoid starting sentences with the word "When".

The conclusions presented do not show the clinical implications of the results found. It is a repetition of the discussion in summary form.

I suggest that the authors rewrite this section with the implications and value of the results found in this study.

Author Response

Dear reviewer and editorial office:

First, I would like to thank you for the trust placed in our group by reviewing and considering our article.

According to the comments received, we have made a series of corrections in our article that I list below, this comments were sent two weeks ago but editor have not included it in the program, sorry for the inconvenience:

The English must be revised by a native speaker and the writing of the entire manuscript must be improved.

We have revised all English throughout the manuscript.

Abstract

The presentation of the abstract needs some clarifications for a better understanding by the reader.

The sections of the abstract; Background, Methods, Results and Conclusions must be presented in italics.

We have changed the beginning of the sections to italic.

Line 10: The aim should still be in the "background" section and should start with. "The aim of the study was to compare...".

We have changed this section as you mention.

Line 19: Quantitative data when expressed in the abstract and throughout the manuscript, I suggest that they be expressed as: 68.26 years (±1.86 years), as well as 23.51 kg/m2 (±4.75 mg/m2). Thus, throughout the abstract and throughout the manuscript with all the variables presented.

We have changed the expression of quantitative variables to your recommendation.

Line 22: p-values ​​should always be presented to three decimal places. Modify it in the abstract and throughout the manuscript.

We have added third decimal to the p-value in all the manuscript.

Introduction

The introduction should have detailed information and provide more exhaustive feedback. Expose and list these different techniques for a better understanding by the reader.

Lines 34-36: Add references that support these sentences that the authors present.

We have included the reference; this reference is the number 1.

Lines 57-60: What techniques define body composition? The authors must present in detail the existence of the theoretical framework.

We show the different techniques in the next lines and its applicability in routine practice. We think a more extensive description of each technique can difficult understanding. Nevertheless, we explain the techniques that it has been used in this study in methods.

Line 61: What does DEXA mean? The first time it occurs in the manuscript, define this acronym. In the same way with all those shown throughout the manuscript.

DEXA means Dual Energy X-ray Absorptiometry. It was written in manuscript.

Lines 64-65: Likewise, I need the authors to support this sentence with a reference. Recently, what do you want to explain with this? References? Relevant studies?

It was mentioned references from Hernandez-Socorro.

Lines 73-75: I´m agreed with the statement indicated, but I consider that it is an assessment more typical of the discussion section. Please change this sentence. Or is it a hypothesis? Think about this sentence.

This sentence pretends to introduce the purpose of our study in validation of a technique as muscle ultrasound. For this reason, it’s included in introduction.

Material and Methods

2.1 Design

Lines 83-92: I suggest that this paragraph be changed and presented in subsection 2.2 Study subjects. This information is not included in the study design. Also, the definition of study variables is preferable to present in later subsections, deleting them from the Design subsection.

We have moved to the suggested sections this part of text.

Lines 90-92: This CEIm is from a Hospital in the Eastern area of ​​Valladolid? It is more specific to present this information from a Hospital Research Ethics Committee with an Institutional Review Board (IRB) instead of a "code number". Declaration of Helsinki; please, describe it according to the last update and the year of creation, for example “The study was in compliance with the Declaration of Helsinki 1964 (last update 2013)”.

Ethical considerations must be rewritten and improved for the manuscript to be considered for publication.

Under Spanish legislation the Institutional Review Boards are “Comité de Ética en Investigación con Medicamentos” or “Medical Research Ethics Committee” of each Health Area. We have changed the topic and we have included code and original positive report from this Committee.

We have also changed the mention to Declaration of Helsinki

2.2 Study subjects

This subsection should be expanded with the considerations outlined above, improving the information on the inclusion and exclusion criteria, enumerating by i) ii) iii) successively.

We have expanded the section and we have included the numbers that you recommend.

2.3 Study variables

The variables are well presented to the reader. Expand the information of the suggested change in subsection 2.1.

We have expanded the section and we have included the numbers that you recommend.

2.4 Data analysis

I suggest that this subsection be renamed “Statistical Analysis”

We have renamed this section as you mention.

Line 150: The sentence "with an official license from the University of Valladolid" is not required. Delete it.

We have deleted this sentence.

Results

This section should be improved with the considerations listed above in abstract and below as follows;

First, lines 158-160 should be removed as they are the template hints and should not be shown to reviewers.

Sorry for the mistake, we have forgot delete this sentence from the template, these sentence has been deleted.

3.1. Sample Characteristics

Instead of naming this section as 3.1 Sample Characteristics, I suggest it be renamed as 3.1 Subject Characteristics

We have renamed this section.

If the authors have Figure 1 in color, they can present it, since it offers a better visualization of the results for the reader.

We have changed figure 1 to color display.

Lines 167-169: Please, in the significant differences, indicate the p-value of the results as indicated in Table 1, these being presented with three decimal places.

We have added third decimal to the p-value in all the manuscript.

3.2 Correlation between body composition assessment techniques

Lines 176-178: It is not necessary for the authors to explain again the analysis performed and presented in Table 2. Rewrite this sentence. It is confusing to the reader.

We have rewritten the sentence to not explain again the analysis.

In Table 2, some results are presented in italics and others are not. Please unify the presentation. p-values, with three decimal places.

The italic representation is for significant values to facilitate lector the localization of these parameters. We have completed to three decimal places the p-values.

3.2.1 Anthropometry

Lines 190-194: Indicate in this paragraph the values ​​p and r and delete them from the figure. In the text, indicate figure 2a and 2b where appropriate, to clarify the presentation of the results.

We have done it as you ask.

Line 195: Please, remove this sentence.

We have removed this sentence.

3.2.2 Body Composition

Lines 199-202: Indicate the p and r values ​​in the text and delete them from the figures. Indicate the results where they correspond as figure 3a, figure 3b, figure 3c... to facilitate the reader's reading.

Lines 207-210: Same considerations as lines 199-202.

We have deleted r and p from the figures and included in text.

3.2.3 Handgrip Strength

Lines 215-217: Same considerations as above. These sentences are difficult to understand. Are there correlations in both MARA and MCR along with muscle/fat mass index? Re-write this sentence.

We have deleted r and p from the figures and included in text and we have rewritten the sentence.

3.3 Diagnosis of malnutrition with GLIM

Don't start most sentences with the word "When". It is not aesthetic in English language. Re-write the beginning of the sentences.

We have rewritten most of the sentences to delete When.

Table 3. p-values, please with three decimals.

We have added three decimals in all variables.

3.4. Diagnosis of sarcopenia

Line 238: Explain the meaning of EWGSOP2, since it appears numerous times throughout the manuscript.

We have included the meaning of EWSOP2 (revised European Working Group on Sarcopenia in Older Patient) criteria in methods.

Change the word criteria instead of "criteria"

Table 4: p-values, please with three decimals.

We have added three decimals in all variables.

Discussion

I recommend the authors to rewrite this section for a better understanding. It is poor in meaning that you want to convey.

We have rewrite most of the discussion with your comments and other reviewers comments to a better understanding to readers.

Avoid starting many of the sentences with the word "When". Use synonyms so as not to be repetitive.

We have changed all the sentences to be less repetitive in these expressions.

I recommend the authors to rewrite this section for a better understanding. It is poor in meaning that you want to convey. I suggest conducting a deeper systematic review to substantiate the claims in this section.

We have added more references, but the expertise in this area (ultrasonography in malnutrition) is limited yet.

Lines 271: The Sanchez-Torralvo and Gort-VanDijk studies should be referenced just after the name, for a better understanding of the comparisons made by the authors. You must re-word the sentences between lines 270-274 to organize this information.

We have added the references and organized the information as you suggest.

Lines 275-277: I suggest re-wording this sentence because the meaning of the Contreras-Bolivar study is not understood; "in total and in differences according to sex"?? What does the author mean with this sentence?

We want to say that our sample showed similar values for handgrip strength that those of Contreras-Bolivar series in total sample and stratified by sex.

Lines 292-294: Is the main objective of the study necessary to explain it in the middle of the Discussion section? Please consider deleting this sentence or moving it around.

You are right. Perhaps, for a better understanding it is not necessary to repeat the main objective in the middle of discussion. We have deleted it.

Line 298: In what "most studies"? Please add references to support this statement.

We have added reference to study of NHANES by Santos et al, and we have added a recent review in oncological population by Aleixo et al.

Lines 295-306: This paragraph is devoid of references that support/disagree with the authors' arguments. Please, improve and re-word this paragraph.

We have added the studies of the preview correction and we have added some references than support our affirmation about BMI, arm and calf circumference.

Line 329: “first” instead of “firs”.

This word has been corrected.

Line 340: “Hanna et al.”, instead of “Hanna eta al,”

We have changed it.

Line 342: Where is the reference for the Catikkas et al. review?

Sorry, we have forgotten adding this reference. We have added the reference to bibliography.

Lines 355-366: On limitations and strengths of the study, please, re-word these paragraphs.

Main limitations or several limitations?

We have deleted the word to avoid confusions.

I would place the strengths before the limitations of the study and number them for a better understanding as i) ii) iii)... so on each strength and limitation.

We have numbered strengths and limitations as you suggest to a better understanding.

Conclusions

Avoid starting sentences with the word "When".

We have deleted these sentences.

The conclusions presented do not show the clinical implications of the results found. It is a repetition of the discussion in summary form.

I suggest that the authors rewrite this section with the implications and value of the results found in this study.

We have rewrite conclusions to show the most important results of the study, but we have added some paragraph to explain implications of this results on routine clinical practice.

Reviewer 4 Report

This nice paper shows the advantage of using ultrasonography, a relatively cheap and quick technique, in assessing cancer-associated malnutrition. I have the following remarks:

  • Ultrasonography has a long learning curve, with quite some intra- and inter-operator variability. The authors should propose how to cope with this.
  • Although many studies show a correlation between phase angle and malnutrition, more recent ones do not, and the authors should comment on this.
  • I would propose to add a small paragraph on cancer-associated cachexia, the consequence of malnutrition in advanced patients, in the Introduction.

Author Response

Dear reviewers and editorial office:

First, I would like to thank you for the trust placed in our group by reviewing and considering our article.

According to the comments received, we have made a series of corrections in our article that I list below:

This nice paper shows the advantage of using ultrasonography, a relatively cheap and quick technique, in assessing cancer-associated malnutrition. I have the following remarks:

  • Ultrasonography has a long learning curve, with quite some intra- and inter-operator variability. The authors should propose how to cope with this.
    • We have tried to reduce the variability with special formation of the person who made the ultrasonography, standardization of measure points and the use of ultrasonogaphy by the same person. It was written in line 147-150: “The ultrasonography was made by the same person who was formed in this technique previously. The areas of measurement were standardized. These actions were made in order to reduce variability intra- and interoperator”.
  • Although many studies show a correlation between phase angle and malnutrition, more recent ones do not, and the authors should comment on this.
    • Thank you for the appointment. We have considered this condition in discussion now. In lines 358-367: “Phase angle has shown a relationship with poorer outcomes in in cancer patients as in study of Axelsson et al in head and neck cancer [28] or Paiva et al in patients with cancer who receiving chemotherapy [29]. Nevertheless, recent studies have shown that is not possible use phase angle as an accurate indicator of malnutrition [30]. In our study we didn’t find differences in phase angle between patients with malnutrition or no malnutrition, or between patients with sarcopenia or no sarcopenia. This could be related with changes in corporal water related with inflammatory state in this patients, or with the fact that all patients were in risk of malnutrition when they are referred to Clinical Nutrition Unit.”
  • I would propose to add a small paragraph on cancer-associated cachexia, the consequence of malnutrition in advanced patients, in the Introduction.
    • We have added a paragraph about cachexia in introduction. Line 58-65: “In cancer patients the caquexia is a multifactorial syndrome that involves multiple factors (inflammatory, reduced intake, treatment damage…) and conditions a continuous loss of muscle mass. This syndrome is characterized by a loss of 5% of weight in last 6 months, body mass index less than 20 kg/m2 and any weight loss more than 2%; or appendicular skeletal mass index compatible with sarcopenia and any weight loss more than 2%. The presence of this disease is related with more complications and poorer outcomes in cancer patients. Body composition and the detection of loss of muscle mass is important to early diagnose of this entity[6]”

Round 2

Reviewer 3 Report

Considerations for improvement are not shown in the manuscript. 

Author Response

(The authors gave the same response as above.)
